# Strategy for the Removal of Satellite Bacteria from the Cultivated Diatom

**Yulia Zakharova *, Artem Marchenkov, Nadezhda Volokitina, Aleksey Morozov**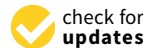**, Yelena Likhoshway and Mikhail Grachev**

Department of Cell Ultrastructure, Limnological Institute, Siberian Branch of the Russian Academy of Sciences, Irkutsk 664033, Russia; marchenkov.am@gmail.com (A.M.); nshelest@mail.ru (N.V.); morozov@lin.irk.ru (A.M.); likhoshway@mail.ru (Y.L.); grachev@lin.irk.ru (M.G.)
* Correspondence: julia.zakharova@gmail.com

**Abstract:** Multiple ecological and genetic studies of diatom algae require an axenic culture. However, algae-associated bacterial biofilms often form in diatom-produced mucus, both during creation of monoclonal cultures from single cells and during biomass growth, and they may be difficult to remove. In this work, we describe a protocol for removing associated bacteria from a monoclonal culture of *Ulnaria danica* isolated from Lake Baikal. The axenization strategy involves selecting the latent phase of diatom growth, multiple washes to remove extracellular polymeric substances and bacterial cells, filter deposition, and treatment with antibiotics that are not toxic for diatoms. The absence of bacteria during these stages was controlled by light microscopy with Alcian blue staining for mucus, epifluorescent microscopy with DAPI (4′,6-diamino-2-phenylindole) staining for bacterial DNA, and scanning electron microscopy of the diatom cell surface. High-throughput sequencing of a 16S rRNA fragment, amplified with universal bacterial primers, from total DNA of a final culture failed to detect any bacterial contamination, confirming successful axenization. A detailed comparative description of all stages of our protocol may prove useful in developing axenic cultures of other diatoms for various ecological and genetic studies.

**Keywords:** xenic cultivating diatom; algal–bacterial association; extracellular polymeric substation; Alcian blue staining; 16S rRNA gene sequencing; axenic culture

## 1. Introduction

Diatoms contribute an estimated 20% to annual C fixation [1]. Some of this carbon is used to produce extracellular polymeric substance (EPS), [2] mostly comprised of polysaccharides [3–5], which is excreted by diatom cells for various purposes and serves as an ecological niche for certain heterotrophic bacteria [6]. Experimental studies of algobacterial interactions and research on the role of these interactions in carbon cycle, as well as other works on microbial ecology, require axenic cultures of diatoms, which are normally associated with bacteria in natural populations.

During the last decades, whole genome studies have gained prominence in diatom ecology. They are able to provide fundamentally novel information about water ecosystems and their functionality [7]. Since the evolutionary history of diatoms involves multiple endosymbioses and their associated genetic transfers, as well as horizontal gene transfers from bacteria [8,9], it is important to use axenic cultures for genomic [8,10–12] and transcriptomic [13–17] studies.

There is a range of methods, which are usually used in combination, to remove bacteria from microalgal cultures. Among the most common are pipetting or washing [18] and treatment with ultrasound [19–22], detergents [23] and antibiotics [24–30], sieving and washing resting stage cells [31]. Purifying existing laboratory monocultures is usually harder than producing axenic cultures from

natural samples. This is because long-term culturing leads to the accumulation of EPS, which is colonized by bacteria. These, in turn, form dense biofilms closely associated with diatom cells [32–34]. Antibiotic treatment is often ineffective because biofilm-forming bacteria show higher antibiotic resistance than their natural planktonic counterparts [35,36]. Furthermore, effectiveness of axenization methods varies from diatom to diatom and from bacterium to bacterium. Raphid pennate diatoms of genus *Ulnaria* excrete large amounts of EPS through their pore fields during culturing, often leading to colony formation, which is not normally observed in these species [37,38]. Heterotrophic bacteria closely associated with diatom algae include multiple phylotypes, particularly α-, β-, γ-Proteobacteria, Bacteroidetes, and Actinobacteria [33,37,39,40].

In this work, we compare various purification methods and propose a protocol for axenizing a monoculture of cosmopolitan freshwater planktonic pennate diatom *Ulnaria danica* (Kützing) Compère and Bukhtiyarova. This protocol involves the following: (1) microscopic identification of bacterial biofilms in various stages of biomass production and culture axenization; (2) multiple washes and filter depositions to remove EPS from diatom cells; (3) antibiotic selection; and (4) axenity verification via high-throughput sequencing of 16S rRNA amplicon.

## 2. Materials and Methods

### 2.1. Isolation and Cultivation of Diatom Strains

*Ulnaria danica*, strain BZ251, was sampled in June 2017 from the photic layer of Barguzin Bay, Lake Baikal (53°27′245″ N; 108°44′387″ E) with a surface water temperature of 2.87 °C. Phytoplankton samples were taken with the Apstein plankton net and placed in sterile plastic flasks with sterile DM medium [41]. Individual cells were isolated in the laboratory by micropipetting under an Axiovert inverted light microscope, placed into a 96-well plate with DM, and grown in a mini-incubator [42] at 10 °C and 16 µmol/m$^2$ s light intensity with a 12:12 day–night cycle. When wells reached a population of ~10$^3$ cells, the contents were moved to 100 mL Erlenmeyer flasks for further biomass growth. Cultures were reinoculated once a month. Each manipulation was performed under sterile conditions. Diatom abundance was calculated as previously described [37]. Strain BZ251 was identified as *Ulnaria Danica* based on its morphological features and a comparison of rbcL sequence with those available in GenBank, as documented previously [38].

### 2.2. Purification of Planktonic Diatom Ulnaria danica

The axenization strategy, based on the protocol [28], involves two stages: mechanical removal of bacteria via wash on filters and antibiotic treatment. The first stage was tested on cultures in different growth stages. In the first case, 50 mL of exponentially growing xenic culture (after 18 days of incubation; density: $3.5 \times 10^4$ cells/mL; the cell density doubled once in two days) was deposited on a 5 µm polycarbonate membrane filter (REATREC-Filter, Russia) using a vacuum pump (Sartorius, Germany). Cells were washed five times with sterile DM right on the filter. Then, they were resuspended into a flask with DM and treated with 20 µg/mL Triton X-100 detergent (Fluka, USA) for 30 s. After that, cells were immediately deposited on a similar filter and again washed five times with DM.

In the second case, the xenic culture on the latent growth stage (after 6 days of incubation; density: $2.5 \times 10^3$ cells/mL; the cell density doubled once in three days) was deposited on a filter as described above, moved to a flask with 50 mL of sterile DM, and carefully mixed by rolling the flask in hands for 5 min at 18 °C. This filtering and resuspension procedure was repeated six times.

For the antibiotic treatment stage, we used 5 µg/mL ciprofloxacin, 100 µg/mL kanamycin and a combination of the two (5/100 µg/mL). Cell cultures growing on DM were incubated with antibiotics for 18 and 36 h under usual culturing conditions. After treatment, cells were deposited on the filter, washed 5–6 times with DM (similarly to the previous stage) and moved to 100 mL flasks with 50 mL of DM.

### 2.3. Verification of Diatom Culture Axenity

For a primary estimate of bacterial contamination, we used phase contrast microscopy and epifluorescent microscopy. An amount of 1 mL of culture was fixed with 1% glutaraldehyde (Sigma, USA) for future staining with DAPI (4′,6-diamino-2-phenylindole) and studied on an Axiovert 200 inverted light microscope using HBO 50W/AC ASRAM ultraviolet lamp at an excitation wavelength of 365 nm, as described in a previous work [28]. Microphotographs were produced using a PIXERA Penguin 600CL camera and AXIOSET software. At the same time, monoclonal *U. danica* BZ251 cultures were inoculated on Petri dishes with various media (LB, FPA/10, DA) to grow any bacteria present, as described in a previous work [37].

To control for bacterial biofilms on diatom cell surfaces, we stained acidic polysaccharides in the mucus. Cell suspension (1 mL, triplicate) was fixed with 2.5% glutaraldehyde solution, stained with Alcian blue at pH 2.5 for 30 min and washed in sterile water. A droplet of stained cell suspension was placed on the object plate, dried on air and examined at $10^3$ magnification using an Axiostar plus optic microscope (Zeiss, Germany). To study EPS via scanning electron microscopy, the diatom culture (1 mL) was fixed for 40 min with 2.5% glutaraldehyde solution and deposited on a filter. The fixed cells were washed with 0.1 M phosphate buffer (pH 7.4) and dehydrated in a series of ethanol solutions (30%, 50%, 70%, 100%) for 5 min in each. After dehydration, the filter was fixed to an aluminum SEM plate, coated with gold using an SCD 004 vacuum coating system, and examined on Quanta 200 scanning electron microscope (FEI Company, USA).

To confirm the complete absence of bacteria, we attempted amplifying bacterial 16S from the culture's total DNA. To produce DNA, algae were deposited on a 0.2 µm polycarbonate filter (Millipore, Ireland), and DNA was isolated from wet sediment, as previously documented [43]. The V4 variable region of the 16S rRNA gene was amplified using universal bacterial primers: F515 5′-GTGCCAGCMGCCGCGGTAA-3′ and R806 5′-GGACTACVSGGGTATCTAAT-3′ with Illumina linkers [44]. Amplification was performed using Phusion Hot Start II High-Fidelity DNA polymerase (Thermo Fisher Scientific). The PCR mixture consisted of 1× Phusion buffer; 1 unit of Phusion polymerase; 0.2 mM equimolar dNTP mixture; 1.5 mM $Mg^{2+}$ and 0.2 µM of each primer; and 10 ng of DNA matrix. The reaction was performed at the following temperature profile: 94 °C, hot start—30 s; 29 amplification cycles (94 °C—30 s, 50 °C—30 s, 72 °C—30 s); and 72 °C, finishing—3 min. The PCR product was purified with AM Pure XP magnetic particles (Beckman Coulter) according to the manufacturer's protocol. Libraries were sequenced on Illumina Miseq with sequencing kit MiSeq® Reagent Kit v3 (2 × 300 b). Amplification of PCR products and sequencing of the libraries was carried out in the Core Centrum "Genomic Technologies, Proteomics and Cell Biology" in ARRIAM (All-Russia Research Institute for Agricultural Microbiology, Russia).

Taxonomic composition of libraries produced from the hypothetically axenic culture was estimated using mothur 1.44.1 [45] according to the standard operating procedure. SILVA seed database release 138, provided by mothur developers, was used for alignment and taxonomic assignment. The sequence data were submitted to the Sequence Read Archive database (https://www.ncbi.nlm.nih.gov/sra/) of the National Center for Biotechnology Information under accession number PRJNA657955. All computations were performed on the HPC cluster "Akademik V.M. Matrosov" of the Irkutsk Supercomputer Center SB RAS. Monoclonal cultures of *U. danica* BZ251, in which bacteria were neither observed microscopically, nor grown on bacterial media, nor detected via 16S sequencing of total DNA, were considered axenic.

### 3. Results and Discussion

### 3.1. Removal of Associated Bacteria from the Diatom Monoclonal Culture

Our previously published protocol [28] was used to produce three axenic strains of a planktonic pennate diatom *S. acus* subsp. *radians* (Kutz.) Skabitsch (strains G9, A6, A280) for genetic and molecular biology work [12,17,43,46,47]. With modifications, it was applied to *Thalassiosira pseudonana* (Hustedt)

Hasle and Heimdal and *Pseudo-nitzschia multiseries* (Hasle) Hasle [39,48], *Navicula phyllepta* Kutzing [49], *Asterionellopsis glacialis* (Castracane) Round, *Nitzschia longissima* (Brébisson) Ralfs [40], and *Thalassiosira rotula* Meunier [50] for the studies of algobacterial interactions.

Initially, we attempted to axenize *U. danica* strain BZ251 according to this protocol. A monoclonal culture was taken from LIN SB RAS diatom collection and grown as described in the Materials and Methods section. After filtering, detergent and antibiotic treatment, the culture was washed on filters yet again, and individual cells were placed into the wells of a 96-well plate with DM. After culturing for 12 days, each monoclonal line (at least 90) was studied using epifluorescent microscopy to identify bacterial contamination. The lines without bacteria were inoculated in 50 mL sterile DM, cultured for 10 days (until they reached exponential stage) and studied under the microscope with DAPI staining. During this latter microscopy check, all diatom cultures were found to be contaminated (Figure 1).

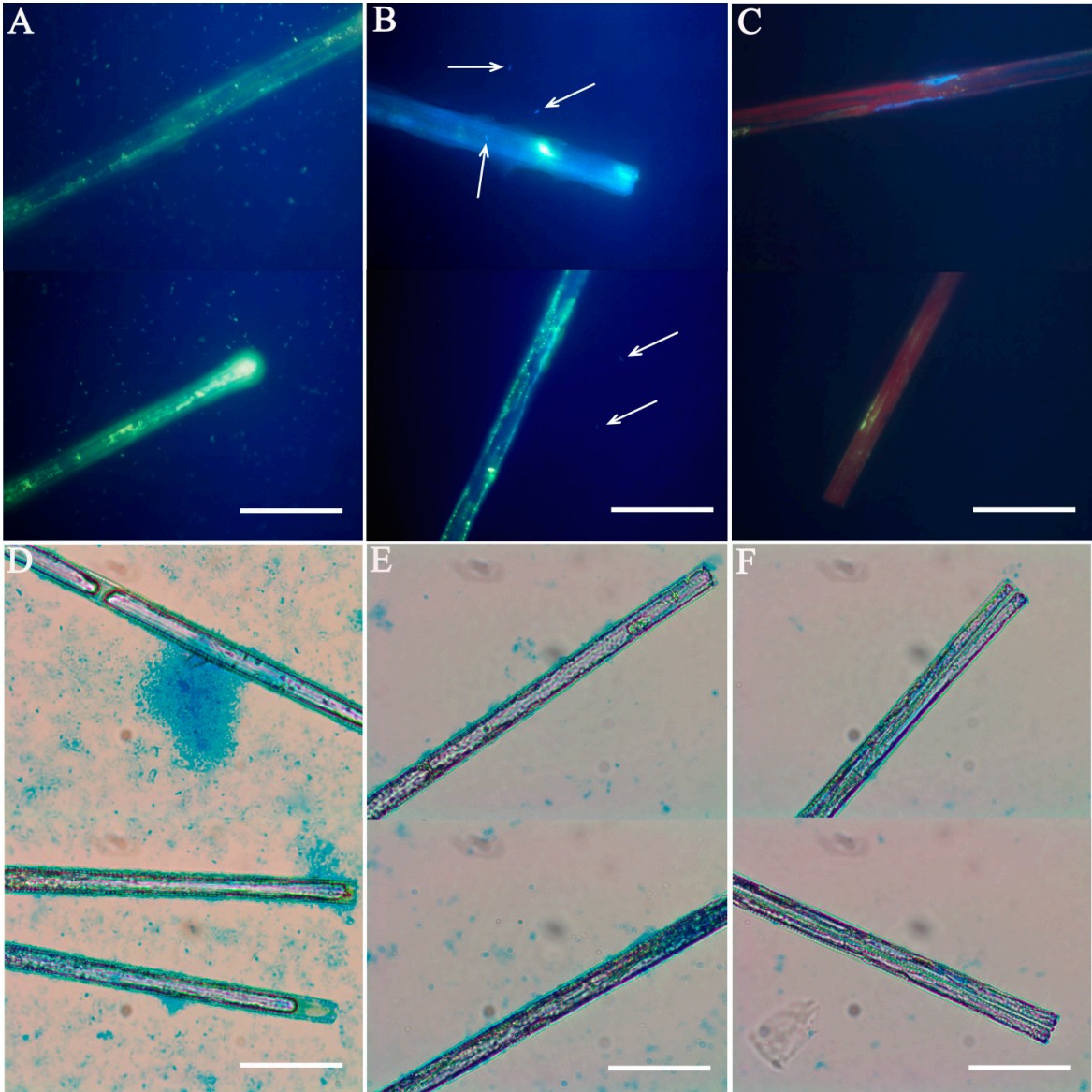

**Figure 1.** Microscopic validation of diatom *Ulnaria danica*. (**A**–**C**) DNA staining: (**A**) xenic culture; (**B**) culture after filtering and washing; (**C**) axenic culture. (**D**–**F**) extracellular polymeric substance (EPS) staining: (**D**) exponential phase of growth; (**E**) latent phase of growth; (**F**) after washing. Bacteria are marked by arrows. Scale bars: 10 μm.

After several failed attempts to produce the axenic cultures using an existing protocol, we attempted to use additional antibiotics. Using a broad range of antibiotics (such as gentamycin, streptomycin, chloramphenicol, and penicillin G) with long treatments and high concentrations is a major method of axenization [22,27,50], but the antibiotics can be toxic for some diatoms [28,49,51,52]. Using kanamycin (100 μg/mL) and a ciprofloxacin/kanamycin cocktail (5/100 μg/mL) for 18 or 36 h [49] after washing failed to remove the bacteria. Furthermore, kanamycin proved to be toxic for *U. danica*, since incubation with this antibiotic arrested culture growth. We observed the culture growth daily for 14 days, but the doubling of cell density was only once on the second day after treatment with kanamycin; then, the growth stopped.

To improve the detection of bacterial contamination on early stages of axenization, we extended the microscopic validation protocol, using Alcian blue to stain polysaccharides for light microscopy, as well as SEM examination. The polysaccharide staining revealed that the EPS layer remains present after the washes (Figure 1); SEM detected both individual bacteria and bacterial biofilms attached to the diatom cells (Figure 2D). Usually, the attached EPS layer is removed from benthic diatoms by centrifugation and soft ultrasound treatment [19,22,27,49,52]. The latter is not applicable to araphid pennates because ultrasound treatment destroys the chloroplasts and drastically reduces cell viability, as shown for *S. acus* previously [28]. Centrifugation is also harmful for *U. danica* because its narrow long cells are easily broken and rendered unviable at higher centrifugation speeds. Detergent exposure followed by filter deposition did remove EPS and decreased bacterial counts, but it also harmed a majority of diatom cells, with more than 80% unable to divide after the treatment. In comparison, multiple washes without detergent treatment reduced bacterial contamination almost by an order of magnitude without an adverse effect on diatom viability.

It has been shown that the EPS production rate in *Navicula salinarum* Grunov and *Cylindrotheca closterium* (Ehrenberg) Reimann and J.C.Lewin was highest during the passage from the exponential to the stationary growth phase [26]. Authors reported that the most effective extraction of attached EPS was achieved by incubating cells in water for 1 h at 30 °C. After water extraction, cells remained alive, while microscopy of Alcian blue-stained pellets showed the complete removal of EPS. With these results in mind, we started the axenization of xenic monoclonal cultures of *U. danica* after 7 days of culturing, during the latent growth phase at $5.0 \times 10^5$ cells/mL. At this time, cell development is still slowed down, and so is EPS production; thus, the bacterial population is also relatively small.

Diatom cells were deposited on a filter, resuspended in sterile DM and carefully mixed by manually shaking the flask; this procedure was repeated six times (Figure 3).

As a result, wash efficiency reached 98% (Table 1).

**Table 1.** Total number of bacteria associated with *U. danica* before and after physico-chemical and antibiotic treatments.

| Treatment | Exponential Phase of Growth (cells/ mL) | Latent Phase of Growth (cells/ mL) |
|---|---|---|
| Initial algal suspension before treatment | $1.0 \times 10^7 \pm 2.9 \times 10^5$ | $5.0 \times 10^5 \pm 4.8 \times 10^4$ |
| Physico-chemical treatment | | |
| Filtering | $9.0 \times 10^5 \pm 7.0 \times 10^4$ | $4.3 \times 10^4 \pm 2.2 \times 10^3$ |
| Triton X-100 *+ filtering | $2.1 \times 10^5 \pm 4.3 \times 10^3$ | not used |
| Washing EPS | not used | $1.0 \times 10^4 \pm 10^3$ |
| Antibiotic treatment after filtering | | |
| Ciprofloxacin | $4.7 \times 10^4 \pm 4.4 \times 10^3$ | no bacteria detected |
| Kanamycin * | $1.5 \times 10^5 \pm 2.2 \times 10^3$ | not used |
| Ciprofloxacin + Kanamycin * | $1.0 \times 10^5 \pm 2.2 \times 10^3$ | not used |

* Detergent and antibiotic toxic for *U. danica*.

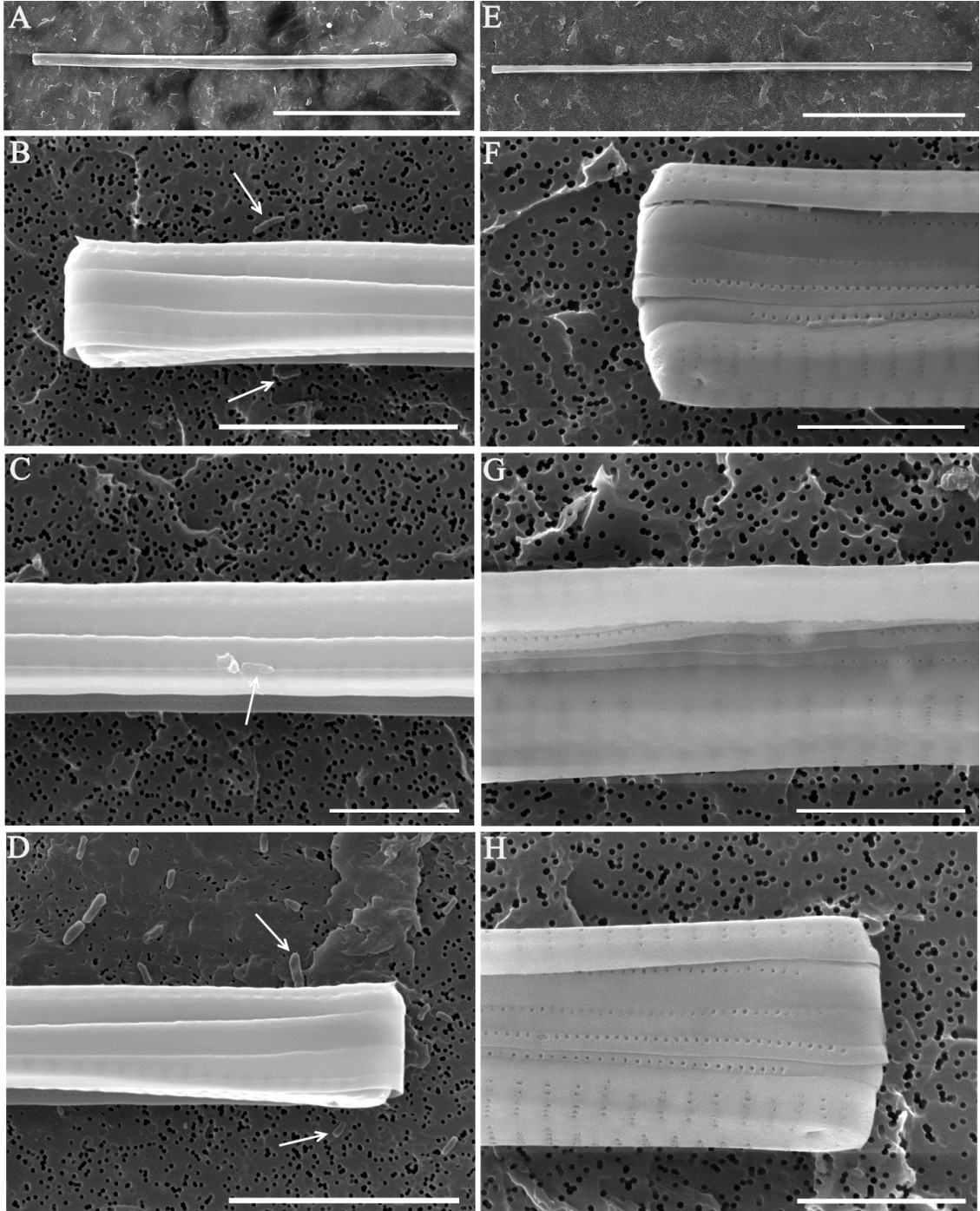

**Figure 2.** SEM validation of diatom *U. danica*. (**A–D**) Xenic culture after filtering and washing; bacteria are marked by arrows. (**E–H**) Axenic culture. Scale bars: A, E—100 μm; B, D—10 μm; C, F, G, H—5 μm.

This approach has allowed us to overcome the issues caused by excessive EPS production during stationary growth phase [53,54]. In addition, multiple resuspensions and filtrations increased the method efficiency by breaking down biofilms and thus decreasing antibiotic resistance. Antibiotic treatments ware performed as described before [26].

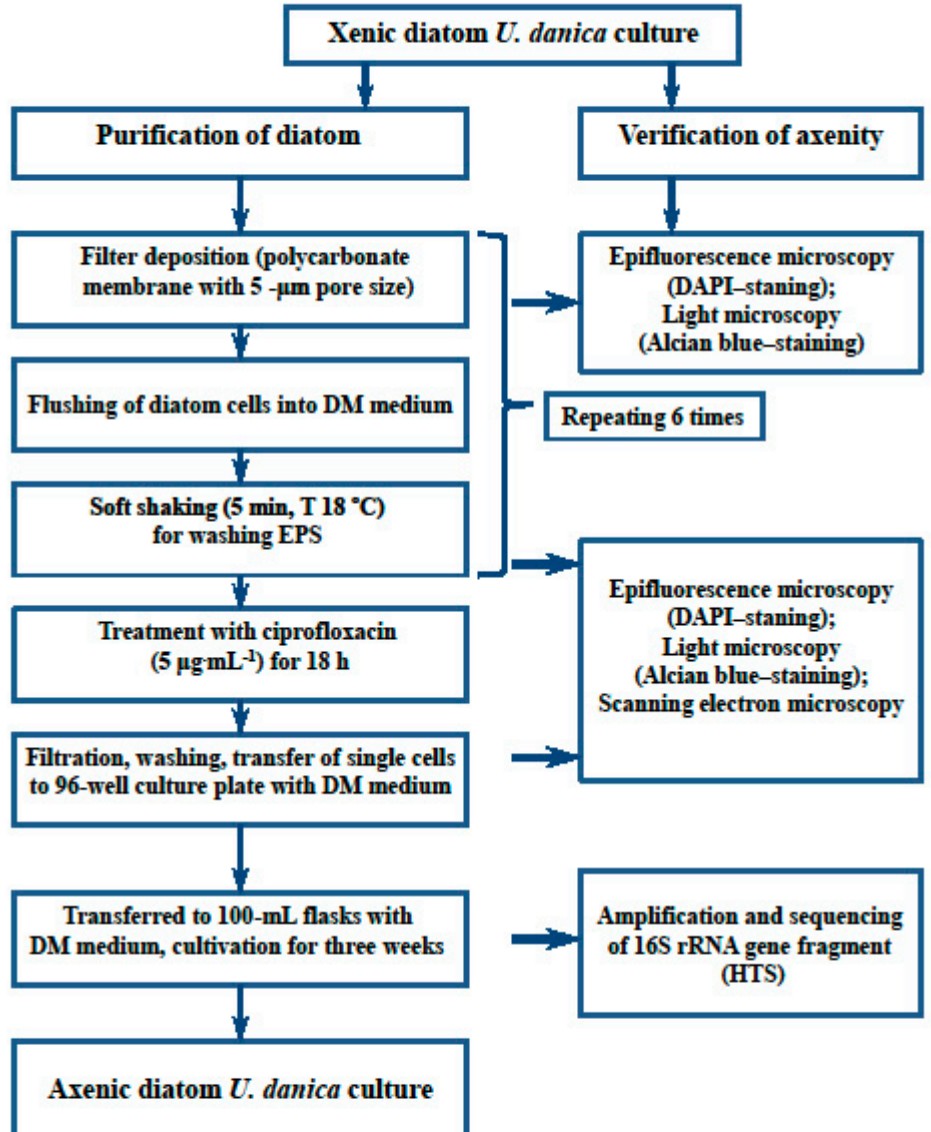

**Figure 3.** Main steps taken during the purification of the culture of planktonic diatom *U. danica* from Lake Baikal.

## 3.2. Verification of Axenity

During each purification stage, we performed epifluorescent microscopy of DAPI-stained preparations. It was shown (Table 1) that both DM washes and antibiotic treatments decrease bacterial counts. Both free-living and diatom-attached bacteria were observed in the culture (Figure 1).

The diatom culture was considered axenic when bacterial DNA fluorescence was not observed. This method is commonly used to control algae axenization [22,27,49,50]. Since large organic or inorganic particles present in the sample can prevent proper focusing on strong magnifications [31,55], we repeated this analysis twice: after initial growth in 96-well plates and in flasks after two or more re-inoculations. In our experience, it is not always easy to simultaneously see diatoms and bacteria in the same field of view because of the drastic difference in cell size (1 μm for bacteria vs. 250 μm for *Ulnaria*). In addition, EPS itself is weakly stained by DAPI, thus masking bacterial cells. Bahulicar and Kroth [5] have even used DAPI staining to study structure and localization of secreted EPS in *Synedra ulna*, describing dense EPS near valve ends and some material along the entire cell. In summary, epifluorescent microscopy requires multiple experiments to detect diatom-associated bacteria and can produce false negatives even under the best circumstances.

Alcian blue dye forms a complex with sulfates and other anionic groups within acidic polysaccharides, staining them blue [56]. It is used to detect various mucous substances produced by living organisms, such as transparent polysaccharide exopolymeric particles [57–59], mucous structures used for substrate attachment [38,60,61], and EPS used by diatoms for colony formation [62]. We used this staining method to measure the degree to which EPS was removed from diatom valves. It was shown that, prior to any treatment, exponential phase cells have denser associated EPS and more free EPS than those in a latent phase (Figure 1D,E). After multiple washes, the volume of EPS was decreasing until it could no longer be detected with Alcian blue.

After washes and the antibiotic treatment, diatom cells were studied with SEM to detect any visible bacteria. Various bacterial morphotypes (such as rods and coccoids) were observed on all stages of axenization; samples (at least six) in which we did not find any bacteria were considered axenic (Figure 2). Most axenization studies only use epifluorescent microscopy to monitor bacterial contamination, but this may produce false negatives, as discussed above. SEM, on the other hand, has the advantage of $\times 10^4$ magnification, making it capable of detecting bacterial cells less than 1 µm in size—in light microscopy, these would be invisible.

After microscopic confirmation of axenity, *U. danica* clones were grown for at least a month to produce biomass. This biomass was used to extract total DNA for 16S rRNA amplification and sequencing. A metabarcoding approach was chosen because of its sensitivity, which is above that of both microscopy and culturing methods. As mentioned before, microscopy may produce false negatives; additionally, it takes a well-trained specialist to detect bacterial cells at low abundance. Molecular methods, on the other hand, are less labor-intensive and possibly cheaper, they do not rely as much on human judgement, and they are more likely to produce false positives (usually due to sample contamination) than false negatives. [22,52]. Although some works on axenization of microalgae do not use molecular methods for detecting bacteria, we consider this step an important part of axenity verification.

The vast majority of classified sequences were assigned to the algal chloroplast. Tens of reads (roughly 0.02% of the library spread across multiple bacterial taxa) were classified as non-organellar prokaryotes; however, in our opinion, these are false positives. Even higher false positive rates of 0.02–0.06% per genus were shown to exist with MiSeq sequencing and SILVA-based mothur classification in a recent method evaluation study [63]. In that work, both misclassifications and detection of bacteria, which had no relatives in the studied community, were observed—the latter explained by cross-contamination between samples in the same sequencing run. Therefore, we conclude that 16S-based analyses agree with microscopic axenity validations, and there is no evidence for the presence of bacteria.

Therefore, our strategy for axenization of a laboratory diatom culture is as follows: start from the culture on the latent growth phase; use multiple washes with sterile DM at 18 °C to remove EPS; treat the culture with antibiotic(s) that are not toxic for the species in question. All stages are controlled microscopically to measure the degree to which EPS and bacteria are removed. Axenity of the resulting culture is confirmed by high-throughput sequencing of 16S rRNA fragments.

**Author Contributions:** Conceptualization, Y.Z. and M.G.; methodology, Y.Z.; data preparation, Y.Z., N.V. and A.M. (Artem Marchenkov); data analysis, Y.Z., A.M. (Artem Marchenkov) and A.M. (Aleksey Morozov); writing—original draft preparation, Y.Z.; writing of the manuscript, Y.Z., A.M. (Artem Marchenkov), A.M. (Aleksey Morozov) and Y.L; writing—review and editing, Y.L. and M.G. All authors have read and agreed to the published version of the manuscript.

**Funding:** This work was supported by the Ministry of Science and Higher Education of Russian Federation projects: no. 0345-2019-0001 (AAAA-A16-116122110059-3) (microscopy); no. 0345-2019-0005 (AAAA-A16-116122110068-5) (cultivation), by Russian Foundation for Basic Research # 17-29-05-030 (DNA extraction and sequencing).

**Acknowledgments:** The study was performed using microscopes of the Instrumental Center "Electron Microscopy" (http://www.lin.irk.ru/copp/) of the Shared Research Facilities for Research "Ultramicroanalysis" LIN SB RAS and using HPC cluster "Akademik V.M. Matrosov" of the Irkutsk Supercomputer Center of SB RAS.

**Conflicts of Interest:** The authors declare no conflict of interest.

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
