# Peer review of "Strategy for the Removal of Satellite Bacteria from the Cultivated Diatom"

_diversity, doi:10.3390/d12100382_

Round 1

Reviewer 1 Report

The authors describe a protocol for removing bacteria associated to a monoclonal culture of U. danica. Starting from a previous published method that they have developed for the production of axenic cultures of S. acus, they implement a protocol for U. danica.

  • The authors state that they have tried to axenize U. danica cultures according to their previously published protocol, used for S. acus; however, this attempt failed. It would be interesting to know then if the protocol described here is fully or in part applicable to other diatoms. 
  • In Section 2.1, please state if manipulation was done under sterile condition and if sterile DM medium was used.
  • How were kanamycin and ciprofloxacin selected for the antibiotic treatment? was an antibiotic susceptibility test performed?
  • There are some typos and other writing mistakes along the text, please correct them (e.g. highlighted text in the Introduction; space missing on the second sentence of par. 2.1)

Author Response

Responses to Reviever 1 to Comments

Point 1: 

Reviewer 2 Report

I read the manuscript with interest since I've working with diatom cultures during decades, and, as authors explain at the introduction, axenic cultures of diatoms are desirable for many research. The use of sequencing for axenity verification is a good point and the manuscript is well written, but I think that the results need some more scientific support. I really missed some information to really see that the culture was purified and for applying the protocol with other diatoms species to make it useful. Since the main aim of such a type of methodological publications is to be used by scientific community, I also consider that authors should improve two questions regarding methods:

  1. They do not clearly explain what do they consider as "latent phase cells", and this make difficult the replicability of the described protocol for other authors, and for other diatoms species. Diatom biology is complex. There are cysts, spores, resting cells, latent cells….and I do not really understand at which type of cell is referring the work.
  2. Most work is based in SEM images, but authors do not give any information about representativemess of the volume (cell density) analyzed with respect the whole culture. It would be desirable some statistical support of this, for instance, the tracing of bacterial density with time.   

I made some comments through the text, in the attached pdf

Author Response

Response to Reviewer 2 to Comments

Reviewer 3 Report

Well written paper and very useful approach for diatomologists to follow with their favorite isolates. 

Definitely takes a lot of effort to get cultures approaching axenicity. 

Would be useful to learn how long the lab cultures stay 'axenic' during routine culture.  i.e. what additional precautions are need to keep the fruits of these labors intact.

Just some minor typos in attached review copy.

Nice work!

Author Response

Response to Reviewer 3 to Comments

Round 2

Reviewer 2 Report

Dear Authors,

The manuscript has been improved. There are some missed errors through the text as follows:

  • page 2-line 39: cells/mL-1 is an uncorrect unit, please correct. 
  • page3-line 9: please change "three replications" by triplicate

After the explanation at page 2-line 38/39 of what the authors do consider "latent growth stage", maybe Lag phase or preexponential growth phase could be more appropiate. Otherwise, cells on latency (benthic diatoms at very late stationary phase, dark, or nutrient limitation) could be misundersanding as I did. 

Author Response

Response to Reviewer 2 Comments

Point 1: cells/mL-1 is ancorrect unit, please correct

Response 1:changed "cells/mL-1" by cells/mL, page 2, lines 32, 39

Point 2: please, change "three replications" by triplicate

Response 2: changed "three replications" by triplicate, page 3, line 9